# Persistent heat waves projected for Middle East and North Africa by the end of the 21st century

**R. Varela** *, **L. Rodríguez-Díaz**, **M. deCastro**

Environmental Physics Laboratory (EphysLab), CIM-UVIGO, Universidade de Vigo, Ourense, Spain

* ruvarela@uvigo.es

**Data Availability Statement:** The daily maximum near-surface air temperature data (Tmax) was retrieved from 13 models of the Coordinated Regional Climate Downscaling Experiment

## Abstract

The duration and intensity of future heat waves are analyzed for 53 cities in the Middle East and the North Africa (MENA) region for the 21$^{st}$ century under two different scenarios (RCP4.5 and RCP8.5). A consistent approach is carried out using data from 13 Regional models within the framework of the Coordinated Regional Climate Downscaling Experiment (CORDEX). By the end of the century, 80% of the most populated MENA cities are expected to be at least 50% of the days under heat wave conditions during the warm season. In addition, the mean and maximum intensity of the heat waves will also increase. Changes in the duration and intensity of heat waves have shown to be negatively correlated. Therefore, the vulnerability of the MENA cities to future heat waves was determined using a cumulative index (CI) that takes into account both duration and intensity. This CI indicates that Middle East and the eastern part of Africa will suffer the most unfavorable temperature conditions in the future. Assuming no intervention trough adaptation/mitigation strategies, these results, together with the particular properties of the MENA region, such as aridity or lack of precipitation, make it likely that the area will be affected by disease or famine.

## 1. Introduction

Today, around 40% of Africans live in cities, which is a percentage considerably lower than in other areas like the European Union, where 75% of the population lives in urban areas as pointed out by Guerreiro et al. (2018) [1]. Even so, according to the McKinsey Global Institute (https://www.mckinsey.com/), over 500 million of Africans live in urban areas and 65 African cities exceed one million population. A megacity is usually defined as a city with a population of more than 10 million people. At present, there are only two megacities in North Africa: Lagos (Nigeria) and Cairo (Egypt). However, the urbanization rate of Africa (~3.5 percent per year) is the fastest in the world and the United Nations (UN) projects that there will be more than 30 megacities in Africa by 2050 [2]. In the Middle East, the urbanization rate increased rapidly during the last three decades of the twentieth century, although it has recently slowed down [3]. According to North Atlantic Treaty Organization (NATO) Strategic Direction South (https://thesouthernhub.org/) there are no megacities in the Middle East. However, projections suggest that cities such as Riyadh, Baghdad or Tehran can become megacities in the

(CORDEX) (https://www.cordex.org/data-access/) for the Africa domain.

**Funding:** This work was partially supported by the European Regional Development Fund (FEDER) under the project "Prevención de Riesgos de Inundaciones y Sequías en la Cuenca Internacional del Miño-Limia" Interreg-Poctep 2014- 2020 (EU. INTERREG-POCTEP 2014-2020, 0034-RISC_ML_6_E) and by Xunta de Galicia under the project ED431C 2017/64-GRC (Grupos de Referencia Competitiva). Both projects were awarded to R.V., L.R-D and M.dC. The funders had no role in study design, data collection and analysis, decision to publish, or preparation of the manuscript. There was no additional external funding received for this study.

**Competing interests:** The authors have declared that no competing interests exist.

near future. The required infrastructure services (e.g., clean water, electricity, transportation, etc) and the agglomeration of population, make cities highly vulnerable to climate change. These challenges are aggravated in Africa where the rapid urban growth generates a continuous increase in the number of people living in slums [4]. According to the World Health Organization more than 70% of the urban population in some sub-Saharan countries lives in slums. Moreover, housing shortages are endemic in most of the Middle East countries, which has forced the population to find alternative solutions like shipping containers or informal settlements on the outskirts of the most populated cities.

Climate change has been one of the major issues studied by the scientific community over the last few decades. Many recent studies have shown that the global temperature has been steadily increasing since the last decades of the 20[th] century [5–7]. Further, a number of recent works have focused on the future evolution of temperature trends, predicting that global warming will continue until the end of the century even under the most favorable scenarios [8–10]. Linked to this rise in temperature, the frequency of occurrence of heat waves has also increased, and they are expected to become more persistent in the future [1, 11–13]. This increase in the occurrence of heat waves has a significant impact on human mortality [14–16], and also important socio-economic implications [17–19].

To this day, the vast majority of the literature on the evolution of the heat waves has focused on some of the most important areas of the world in terms of population, economy or biodiversity [1, 20, 21]. Yet, the countries in these areas are generally better prepared to react and to adapt to the impact of heat waves. In contrast, the Middle East and North Africa region (MENA), represents one of the most vulnerable areas [22–24], with the highest risk of drought, desertification or disease. The importance of conducting studies over the MENA region lies mainly in the potential catastrophic impact on human health [15, 25–27], and the difficulty of the population of these countries to mitigate or adapt to these impacts. Recent studies have focused on the future difficulties that the MENA region will face in terms of water demands, high population growth, sudden flows of refugees and economic development [28–31].

The MENA region covers all African countries north of the equator, as well as the Middle East nations (Fig 1). The study area presents remarkable differences in the growth and income patterns [32]. In the past 20 years, MENA region showed the highest population gross in the world [33]. Besides, the vast majority of MENA are characterized by frequent extreme temperature events (>35˚C) [24, 34] and severe drought in summer [35], leading to large arid zones that may increase the risk of famine and famine-related mortality. Recent studies focused on this region have demonstrated that temperature will continue to increase [36–40] and thus, episodes of unusual extreme temperatures and heat waves are expected to rise in the future [11, 22–23, 41–43].

Based on our knowledge now, the studies focused on heat waves in the MENA region have only been carried out at global scale [39–40, 43, 44] or cover a small number of cities [23, 38]. The relevance of the present work lies in the need to conduct a consistent approach to analyze future changes in the main characteristics of heat waves in the main MENA cities. In this sense, warming rates, extreme temperatures and the duration and intensity of heat waves were analyzed under the RCP4.5 and RCP8.5 scenarios for two future periods 2020–2049 and 2070–2099.

## 2. Data and methods

### 2.1 Maximum temperature

The daily maximum near-surface air temperature data ($T_{max}$) was retrieved from 13 models of the Coordinated Regional Climate Downscaling Experiment (CORDEX) (https://www.cordex.org/data-access/) for the Africa domain (S1 Table). The use of RCMs constitutes an

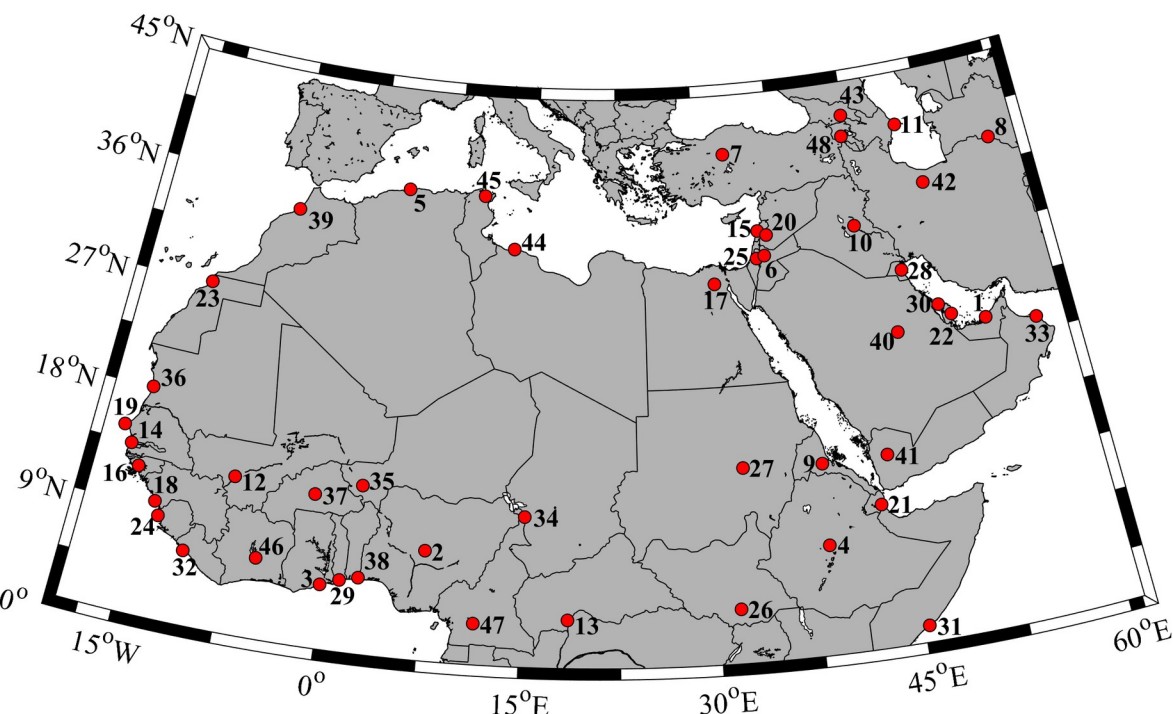

**Fig 1. Area under scope.** Each red spot indicates the coordinates of each capital in alphabetical order: 1. Abidjan (Ivory Coast); 2. Abu Dhabi (Emirates); 3. Abuja (Nigeria); 4. Accra (Ghana); 5. Addis Ababa (Ethiopia); 6. Alexandria (Egypt); 7. Algiers (Algeria); 8. Amman (Jordan); 9. Ankara (Turkey); 10. Ashgabat (Turkmenistan); 11. Asmara (Eritrea); 12. Baghdad (Iraq); 13. Baku (Azerbaijan); 14. Bamako (Mali); 15. Bangui (Central African Republic); 16. Banjul (The Gambia); 17. Beirut (Lebanon); 18. Bissau (Guinea-Bissau); 19. Cairo (Egypt); 20. Conakry (Guinea); 21. Dakar (Senegal); 22. Damascus (Syria); 23. Djibouti (Djibouti); 24. Doha (Qatar); 25. El-Aiun (Western Sahara); 26. Freetown (Sierra Leone); 27. Giza (Egypt); 28. Istanbul (Turkey); 29. Jerusalem (Israel); 30. Juba (South Sudan); 31. Khartoum (Sudan); 32. Kuwait City (Kuwait); 33. Lagos Metropolitan Area (Nigeria); 34. Lome (Togo); 35. Manama (Bahrain); 36. Mogadishu (Somalia); 37. Monrovia (Liberia); 38. Muscat (Oman); 39. N'Djamena (Chad); 40. Niamey (Niger); 41. Nouakchott (Mauritania); 42. Ouagadougou (Burkina Faso); 43. Porto-Novo (Benin); 44. Rabat (Morocco); 45. Riyadh (Saudi Arabia); 46. Sanaa (Yemen); 47. Tehran (Iran); 48. Tbilisi (Georgia); 49. Tripoli (Libya); 50. Tunis (Tunisia); 51. Yamoussoukro (Ivory Coast); 52. Yaounde (Cameroon); 53. Yerevan (Armenia).

improvement from Global Climate Models (GCMs) due to the capability of the formers to reproduce extreme events at finer scales [40, 44–46]. A detailed description of the model validation for the historical period can be found in Russo et al., (2016) [47] and Dosio (2017) [44] who obtained a good agreement between the data from the CORDEX models and historical values for the MENA region.

The study area covers the MENA region, which ranges from 0 to 42.24˚N and from 20˚W to 60.28˚E, with a 0.44˚ × 0.44˚ spatial resolution [48]. Extreme temperatures and heat waves were analyzed for the MENA region, focusing on regions surrounding 53 cities (all capitals and some very populated cities, see Fig 1) over the historical period (1951–2005) and for future projections (2006–2099) under the Radiative Concentration Pathways scenarios, RCP4.5 and RCP8.5 [49]. To this end, we selected the closest grid node to the city coordinates. Then, a bilinear interpolation was performed considering the four grid nodes closest to the city to determine the $T_{max}$.

The set of months with the highest mean maximum temperature considered for each of the 53 cities was selected based on data from https://en.climate-data.org/. Months whose mean maximum temperature surpass the 75th percentile were considered, which guarantees a minimum of three months per city. The set of months considered for each city will be referred as the warm season from now on. The use of this threshold is due to the particular characteristics

of the MENA region. For example, those cities closest to the equator show hardly any seasonality while others show a clear season of warmer temperatures. Thus, the warm season is different for each city.

A multi-model approach was considered to analyze future changes in $T_{max}$ during the warm season. In this sense, $\Delta T_{max} = <T^F_{max} - T^L_{max}>_n$, where $n$ is the number of RCMs (13 models from CORDEX), $T^F_{max}$ refers to the maximum temperature data for each future period (2020–2049, 2070–2099), and $T^L_{max}$ to the maximum temperature for the late 20th century period (1970–1999). The multi-model approach minimizes the bias and errors of every single model. In addition, the consensus criterion [50, 51] which provides significance to $\Delta T_{max}$, was applied to determine the agreement of the 13 models that make up the multi-model. The consensus criterion is based on imposing the two following conditions:

Firstly, $\Delta T_{max}$ were calculated for both the multi-model mean and for each model separately. The first condition is that for every pixel, at least 60% of the models have the same sign as the multi-model mean. In addition, $\Delta T_{max}$ values beyond of the multi-model mean ± 2 standard deviations (SD) were considered as outliers.

Second, the Mann-Whitney (or Wilcoxon rank sum test) nonparametric test was applied at every pixel, both for each model and for the multi-model mean [52]. A 5% significance level was considered at each grid point. To fulfill the second condition, it was imposed that at least 60% of the models that fulfilled the first condition passed the Mann-Whiney-Wilcoxon test.

## 2.2 Heat waves

There are different approaches to define heat waves that differ with respect to the number of consecutive days under extreme conditions (generally two-six consecutive days) [53–55] and the percentile used to consider an extreme condition (generally the 90th or 95th percentile) [56–58]. In addition, there are multiple ways to estimate heat waves, such as climate indices based on minimum and maximum daily temperature, percentile based indices or the recently developed heat wave magnitude index (HWMI) among others [13, 17, 59–62]. In the present work, the parameters used by Guerreiro et al., (2018) [1] to analyze future heat waves in European cities were adopted. The heat wave days (in percentage), which measure the number of days that a city is under heat wave conditions; and the intensity, which measures the temperature in excess during the heat waves.

In the present work, the temperature threshold ($T_{TH}$) was defined as the 95th percentile of the daily maximum temperature centered on a 31-day window [13, 44, 63] over the months of interest for the reference period (1951–2005). Thus, the number of days under extreme temperature conditions are those which exceed the temperature threshold. To consider a heat wave we establish a minimum of three consecutive days under extreme temperature conditions, heat wave days from now on. The number of days under extreme temperature and the heat wave days were expressed as a percentage to facilitate comparison between different locations where the duration of the warm season is different. The warm season, which can differ from location to location, is shown in S2 Table.

In addition, the mean and maximum heat wave intensity were analyzed. In this sense, the mean heat wave intensity for each location was calculated as $\langle I \rangle = <T^{HW}_{max} - T_{TH}>_{n,t}$ where $T^{HW}_{max}$ refers to the maximum temperature of the days under heat wave conditions, $T_{TH}$ to the daily temperature threshold described above, being $n$ the number of RCMs (13 models from CORDEX), and $t$ the heat wave days observed during the warm season. The maximum heat wave intensity, $MI = <\max(T^{HW}_{max} - T_{TH})>_{n,t}$, which is similar to the one used by Guerreiro et al. (2018) [1], was also calculated for every city.

## 3. Results

### Maximum air temperature

The increment in the maximum temperature averaged for the near (2020–2049) and far future (2070–2099) under the scenarios RCP4.5 and RCP8.5 is shown in Fig 2. This increment was calculated with respect to the late 20th century period (1970–1999). Results reveal that the maximum temperature will increase throughout the whole century regardless of the scenario, with a more evident rise for the far future (Fig 2B and 2D) than for the near future (Fig 2A and 2C). In particular, for the near future, a slight increment that ranges from 1 to 2.5˚C for the whole region was observed without important differences between both scenarios. This increase is clearly accentuated towards the end of the century (Fig 2B and 2D) especially under the RCP8.5 scenario with increments between 4 and 7˚C. The highest maximum temperatures (S2 Table) are observed for some countries situated in the Persian Gulf as Saudi Arabia and in the Eastern Mediterranean area as Turkey or Egypt. Apart from the dramatic increase in the maximum temperatures of the MENA region, it should be taken into account that most of the countries in the zone are already among the hottest locations in the world, with large areas prone to aridity. The continuous increase in maximum temperatures is expected to cause severe consequences as a sudden increase in the number of days with extreme temperatures and under heat wave conditions.

### Extreme temperature conditions

The mean percentage of days under extreme temperature conditions for the near and far future under both scenarios is shown in Fig 3 (additional information is provided in S3 Table).

The percentage of days under extreme temperature conditions for the near future is similar under both scenarios (Fig 3A and 3C), albeit slightly higher under the RCP8.5 scenario. When focusing on this scenario the coastal cities of North Africa present, in general, moderate percentages of days under extreme temperatures, varying from 8.9% in El-Aiun (Western Sahara)

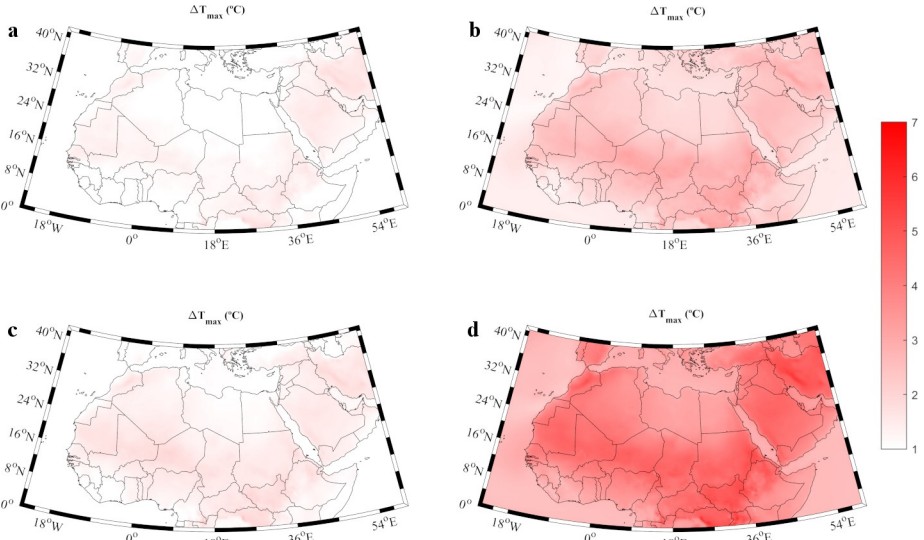

**Fig 2. Increment in the maximum temperature ($\Delta T_{max}$,˚C) during the warm season for: a)** near future (2020–2049) under the RCP4.5 scenario; **b)** far future (2070–2099) under the RCP4.5 scenario; **c)** near future under the RCP8.5 scenario; **d)** far future under the RCP8.5 scenario. Increments were calculated with respect to the reference period (1970–1999). Note that the months under study can vary among locations (S2 Table).

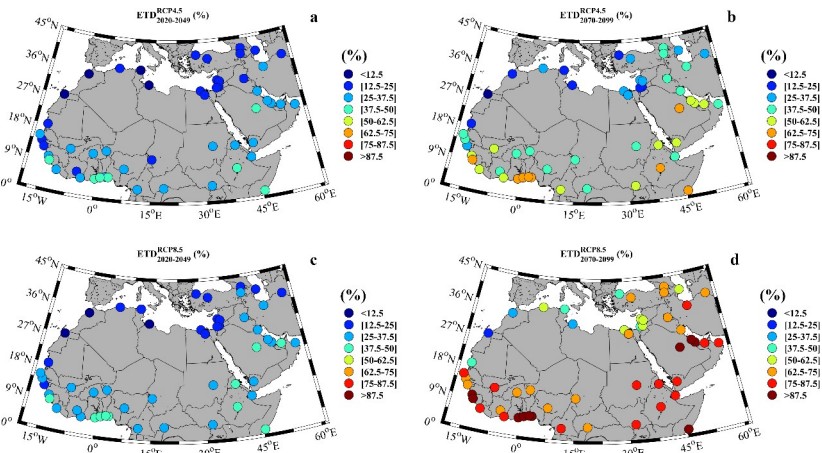

**Fig 3. Percentage of days under extreme temperature conditions for: a)** near future (2020–2049) under the RCP4.5 scenario; **b)** far future (2070–2099) under the RCP4.5 scenario; **c)** near future under the RCP8.5 scenario; **d)** far future under the RCP8.5 scenario. Note that the months can be different for every location (S3 Table).

to 22.0% in Cairo (Egypt), presumably because of the tempering influence of the Mediterranean Sea. In contrast, in Western Africa, the cities located in a coastal fringe close to the Atlantic Ocean (from Dakar to the Gulf of Guinea) show some of the highest percentage of days under extreme temperatures. The cases of Lome (Togo) and Accra (Ghana) are especially striking with values of 44.4% and 47.5%, respectively. A similar behavior occurs in East Africa where Addis Ababa in Ethiopia and Mogadishu in Somalia show values close to 50%. The countries in the eastern part of the Arabian Peninsula or in the Persian Gulf show a similar trend with values exceeding 30% in the entire region (Yemen, Oman, United Arab Emirates, Kuwait, Qatar, Bahrain), with the example of Saudi Arabia exhibiting 49.6% of days that exceed the threshold.

The percentage of extreme days increases markedly in the far future (Fig 3B and 3D), when a majority of cities will double or triple the values obtained for the near future. For example, in the North African coastal city of Rabat or in Tunisia, the percentage of extreme days is 12.1% and 14.2%, respectively for the near future and 25.5% and 44.4%, respectively for the far future under the RCP8.5 scenario. Again, the western margin of Africa is the area most affected, with values above 90% in countries like Guinea, Ghana or Togo under the least favorable greenhouse gas emission scenario. These percentages mean that the projected maximum temperatures will rise to dangerous levels by the end of the century, possibly beyond sustainable limits for life. In effect, nearly every day of the warm season will be an extreme temperature day with potentially disastrous consequences. With the exception of the coastal countries, the great majority of North Africa will be affected by extreme temperatures more than 80% of days. This conclusion also applies for the countries of the eastern Arabic Peninsula where more than 80% of the days will present extreme temperatures, being Saudi Arabia the most extreme case with 94.3% of the days (see S3 Table).

The percentage of heat wave days for the near and far future under both scenarios is shown in Fig 4. Note that a minimum of three consecutive days with extreme temperatures is needed to fulfill the heat wave condition as mentioned above. In this sense, Fig 4 provides additional information compared with Fig 3 that did not clarify whether the extreme temperature days were consecutive or not. For example, 25.5% of the days will be under extreme temperatures conditions at Rabat in the far future under the RCP8.5 scenario (Fig 3D), but only 20.7% of the days will be heat waves days (Fig 4D). This value means that out of the 92 days of the warm

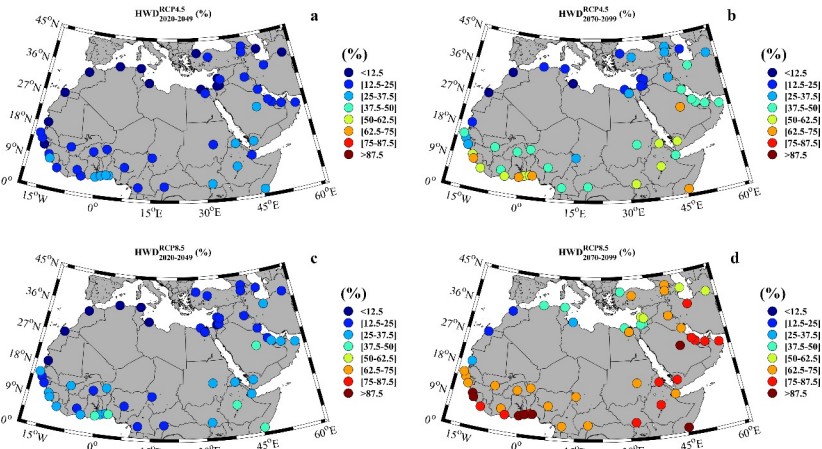

**Fig 4. Percentage of heat wave days for: a)** near future (2020–2049) under the RCP4.5 scenario; **b)** far future (2070–2099) under the RCP4.5 scenario; **c)** near future under the RCP8.5 scenario; **d)** far future under the RCP8.5 scenario. Note that the months under study can vary among locations (S4 Table).

season (July to September), around 18 days will be heat wave days. The percentage of days under heat wave conditions for the near future (Fig 4A and 4C) is similar for both RCP scenarios and much higher than that obtained during the late 20th century (1970–1999) (see S4 Table). When focusing on the near future under the RCP8.5, the cities situated in the Mediterranean Sea margin are less influenced by heat waves, with values exceeding 15% only in Cairo (Egypt). The percentage of heat wave days is higher for the western part of Africa, where values between 20% and 30% are common, with cities such as Accra (Ghana) or Lome (Togo) affected by heat wave conditions nearly 40% of the days. The situation is even more critical in the eastern regions of Africa, where values in countries like Ethiopia or Somalia exceed 40%. In the eastern Arabian countries, the situation is not better, with values around 30% in most locations, except for Saudi Arabia where 44.7% of the days are heat wave days. The situation becomes even more critical for the far future (Fig 4B and 4D), when differences between both RCP scenarios are more marked. All countries except those in the Mediterranean margin (which include the cities of Rabat, Tripoli or Jerusalem) will have a percentage of heat wave days higher than 60%. In average, most of the analyzed MENA cities will be under heat wave conditions during at least 66% of the days of the warm season for both RCP scenarios. Further, in some locations like Accra (Ghana), Lome (Togo), Mogadishu (Somalia) or Riyadh (Saudi Arabia), heat wave conditions will occur for 91.7%, 90.2%, 96.0% and 93.3%, respectively under the RCP8.5 scenario. This means that these locations will actually be under quasi-permanent heat wave conditions. Focusing on the far future, 25% (80%) of the MENA cities will be more than 50% of days under heat wave conditions according to the RCP4.5 (RCP8.5) scenario. Actually, for the latter scenario, approximately 50% of the cities will be more than 70% of days under heat wave conditions. These values are very high especially when compared with those obtained for the late 20th century when the days under heat wave conditions did not exceed, in any case, 3%.

Spatial differences in the percentage of heat waves were evaluated using a clustering method (Fig 5). Two different clusters were obtained for the far future (2070–2099) regardless of the RCP scenario. In general, the highest values were obtained for the southernmost latitudes of Africa (red color) and the lowest (blue color) for the Mediterranean cities in northern latitudes. The cities of the Arabian Peninsula also showed the same behavior as the southernmost latitudes of Africa. The cluster analysis for the near future (2020–2049) can be seen in S1 Fig.

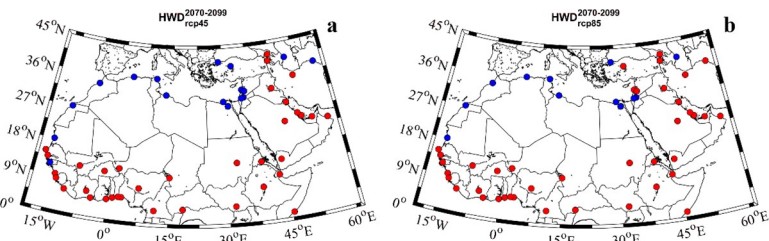

**Fig 5. Cluster analysis of the percentage of heat wave days for the far future (2070–2099) under the a)** RCP4.5 scenario; **b)** RCP8.5 scenario. Red (blue color) indicate high (low) percentages of heat wave days.

## Heat waves intensity

In addition to their duration, heat waves can also be analyzed in terms of their $<I>$ (mean intensity) and MI (maximum intensity). Both parameters refer to the temperature in excess with regard to a certain threshold, as defined above. Every city of the MENA region shows a rise in their $<I>$ and MI of heat waves (see S5 Table). For the late 20th century, the $<I>$ ranges from 0.4°C in Lome to 2.2°C in El-Aiun, whilst the MI ranges from 1.4°C in Accra and Lome to 8.6°C in Alexandria. The $<I>$ for the days under heat wave conditions calculated over the different periods and scenarios is shown in Fig 6. Near future projections display similar results under both scenarios (Fig 6A and 6C), with values ranging from 0.5°C to 1.5°C in the southern part of Africa and Middle East. Higher values ranging from 1.5°C to 3°C were observed for the northern part of the MENA region.

A similar spatial pattern is observed for the far future under the RCP4.5 scenario although, with slightly higher intensities (Fig 6B). A great increase in the $<I>$ is observed for the far future under the RCP8.5 scenario (Fig 6D). In particular, the highest values ($> 3$°C) are observed for the Mediterranean area, Middle East and the eastern part of Africa. On the other hand, the least affected cities are situated in the Gulf of Guinea, where the $<I>$ in cities as Bissau, Conakry or Dakar does not exceed 2°C.

MI shows a similar spatial pattern with lower values south 27°N and higher values at the northernmost part, regardless the scenario and the future period (Fig 7 and S5 Table). In

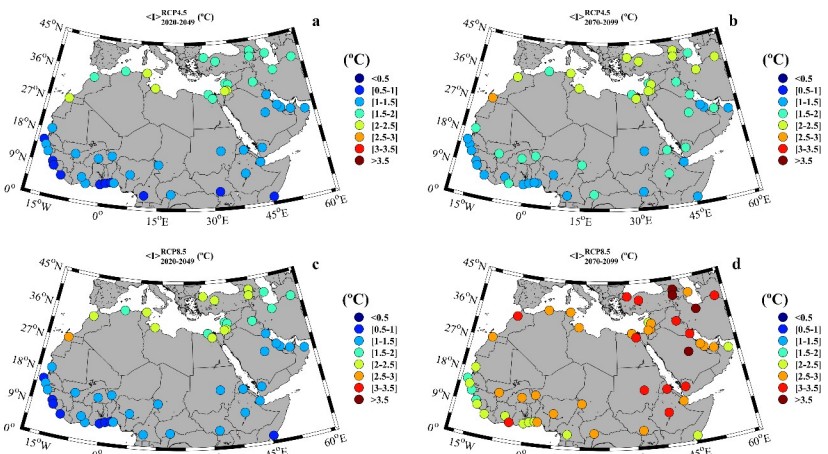

**Fig 6. Mean heat wave intensity (°C) for: a)** near future (2020–2049) under the RCP4.5 scenario; **b)** far future (2070–2099) under the RCP4.5 scenario; **c)** near future under the RCP8.5 scenario; **d)** far future under the RCP8.5 scenario. Note that the months under study can vary among locations (S5 Table).

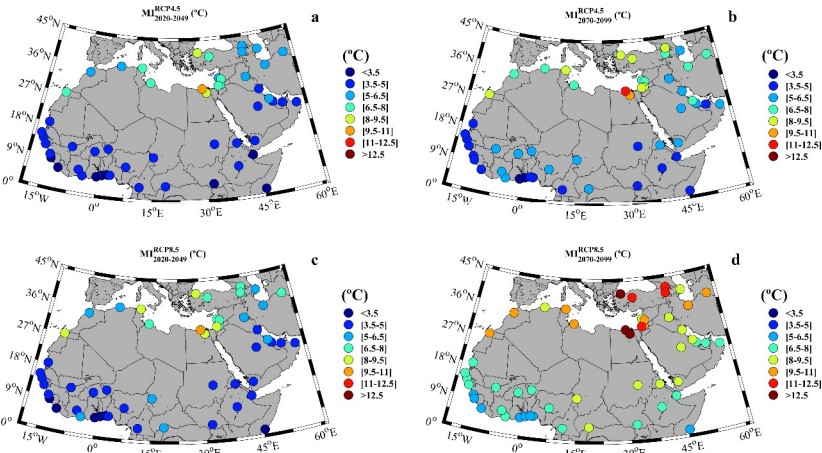

**Fig 7. Maximum heat wave intensity (˚C) for: a)** near future (2020–2049) under the RCP4.5 scenario; **b)** far future (2070–2099) under the RCP4.5 scenario; **c)** near future under the RCP8.5 scenario; **d)** far future under the RCP8.5 scenario. Note that the months under study can vary among locations (S5 Table).

particular, for the far future under the RCP8.5 scenario, IM will exceed 12˚C at some cities as Alexandria, Giza or Istanbul (Fig 7D) and will be lower than 5.5˚C for cities as Accra, Lome, Monrovia and Mogadishu.

The cluster analysis of the <I> and MI of heat waves showed two different behaviors for the far future (Fig 8). Under the RCP4.5 scenario, the highest <I> values were obtained for the northernmost latitudes of Africa and the Middle East (red color) and the lowest for southern latitudes (blue dots). Under the RCP8.5 scenario, those cities situated in the Atlantic Ocean coast showed the lowest values. Regarding the MI, both scenarios displayed similar behavior. Those cities situated on the northernmost (southernmost) latitudes showed the

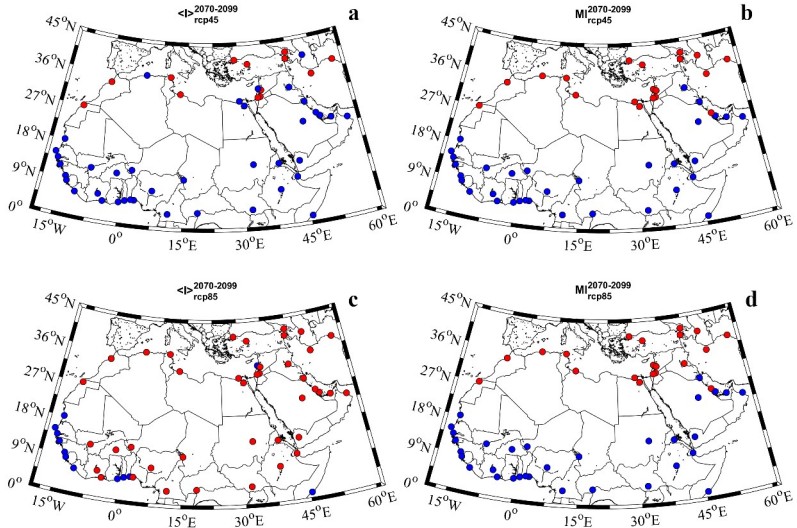

**Fig 8. Cluster analysis of the mean (left panels) and maximum (right panels) heat wave intensity for the far future (2070–2099) under the a, b)** RCP4.5 scenario; **c, d)** RCP8.5 scenario. Red (blue color) indicate high (low) percentages of heat wave days.

highest (lowest) values. The cluster analysis for the near future (2020–2049) can be observed in
S2 Fig.

## 4. Discussion

The increment of temperatures is an accepted fact for the MENA region [36, 44]. Lelieveld
et al., (2016) [43] studied the heat extremes for the 21st century under the RCP4.5 and 8.5 sce-
narios, obtaining temperature increments higher that 5˚C, especially in the Middle East during
the warm season and under the less favorable scenario. Similar results are described in Buc-
chignani et al., (2018) [39] or Ozturk et al., (2018) [40]. Dosio (2017) [44] analyzed projections
of temperature and heat waves for the Africa region by means of a CORDEX ensemble for the
period 2071–2100, being Middle East the area with the maximum expected temperature
increase (up to 6˚C) during the warm season. These results are similar to those presented in
the Fig 2 of the present study. Dosio (2017) [44] also observed heat waves lasting between 60 to
120 days in some areas of the MENA region under RCP8.5 scenario in good agreement with
the present work, where cities in the Arabian Peninsula as Riyadh are projected to suffer
around 80 heat-wave days per year by the end of the 21st century.

As mentioned above, the results in the present study are comparable to previous research
when observed at macroscopic scale. In addition, we have focused on the most populated areas
(the cities), where climatic issues can be aggravated by the agglomeration of people and the
lack of infrastructures to provide the basic services. As far as we know, only a few works have
focused on particular locations. Giugni et al., (2015) [23] observed the impact of climate
change in some African cities: Addis Ababa, Dar Es Salaam, Douala, Ouagadougou and Saint
Louis for the 21st century under the RCP 4.5 and 8.5 scenarios. In agreement with their results,
both Addis Ababa and Ouagadougou showed a dramatic increase in the expected duration of
the heat waves in the present study. Pal and Eltahir (2015) [38] studied the temperature projec-
tions by the end of the 21st century under the RCP8.5 scenario for the southwest Asia which
includes some of the areas of the present work. They observed that cities as Kuwait City,
Riyadh, Doha or Abu Dhabi will suffer more days that exceed the 95th percentile in summer,
with maximum temperatures above 45˚C. Their results are in concordance with those
obtained in the present work, where Kuwait City, Abu Dhabi and, Riyadh are projected to
attain maximum temperatures of 45.5˚C, 43.7˚C and 47.3˚C, respectively.

The regional differences in the MENA region were assessed by means of a clustering
method (Figs 5 and 8). On the one hand, the percentage of heat wave days showed two differ-
ent spatial behaviors for the far future. The southernmost latitudes displayed the highest values
(red color) while the northern latitudes showed the lowest ones (blue color). This distribution
was very similar to that obtained for the percentage of days under extreme temperature condi-
tions (Fig 3). On the other hand, the spatial distribution of the <I> and MI also showed two
different clusters. In general, both <I> and MI displayed the highest (lowest) values for the
northern (southern) latitudes. These results indicate a negative correlation between the inten-
sity and the number of heat wave days. For example, for the near future under the RCP 8.5 sce-
nario, Accra shows 40.1% of the days under heat wave conditions with a <I> of 0.8˚C and a
MI of 3˚C. El-Aiun shows only a 5.7% of the days under heat wave conditions. However, its
<I> is 2.5˚C and its MI is 8.1˚C. Comparing both cities, Accra will suffer more days under
heat wave conditions than El-Aiun, but the intensity of those days will be higher in El-Aiun
than in Accra.

Table 1 summarizes the Spearman correlation between heat wave days and the <I> and
MI. The correlation is clearly negative and significant for both variables and for most of the
periods and scenarios. A similar behavior was found by Guerreiro et al. (2018) [1] when

**Table 1. Spearman correlation (r) between heat wave days (HWD) and intensity (mean (<I>) and maximum (MI) under RCP4.5 and RCP8.5 scenarios for the near and far future.**

| Scenario | Future | HWD *vs* <I> | HWD *vs* MI |
|----------|--------|--------------|-------------|
| RCP45 | Near | -0.78 | -0.80 |
|       | Far | -0.71 | -0.77 |
| RCP85 | Near | -0.74 | -0.76 |
|       | Far | -0.20* | -0.69 |

* marks the correlations with a significance lower than 99.9%.

analyzing the most important European cities, although they only considered the correlation between heat wave days and the MI for the high impact scenario. They also found a negative correlation although a bit lower (around -0.44) than in the present study where the mean correlation (averaging the value of the last column in Table 1) is ~-0.76.

The negative correlation indicates that the cities where heat waves have higher intensity also tend to have shorter duration, and vice versa. Thus, to determine the cities more threatened by severe heat waves in the future, it is necessary to consider both variables. In this sense, a cumulative index (CI) was defined following [64, 65]. This index is calculated as the probability of heat wave days (fraction ∈ [0,1]) times the <I> defined above. Fig 9 shows the CI for the

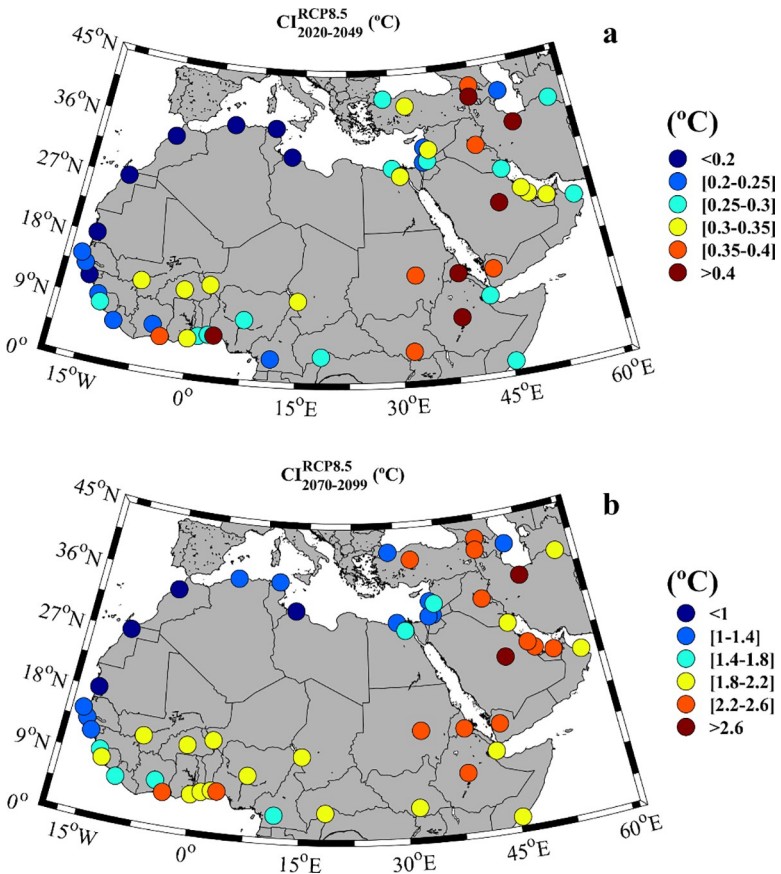

**Fig 9. Cumulative Index (˚C) for a)** the near future (2020–2049) and **b)** the far future (2070–2099) under the RCP8.5 scenario. Note that the months under study can vary among locations (S6 Table).

near (a) and far future (b) calculated under the less favorable scenario (RCP8.5). A different scale was used for each period to enhance the differences among places. Both plots share some common features: i) the lowest CI values are observed for the Mediterranean region and the Atlantic coast till the Guinea Gulf; ii) the highest CI values are observed for the Middle East area and the Eastern part of Africa; iii) intermediate values, alternating high and low values depending on the particular location of the cities, are observed for the most equatorial zone covering from the Gulf of Guinea in the Atlantic Ocean till Somalia in the Indian Ocean. The CI is clearly more intense for the far future (Fig 9B), at least five times higher, than for the near future (Fig 9A). For comparison purposes, the CI was calculated for the top ten European cities (in terms of intensity or duration) analyzed by Guerreiro et al. (2018) [1] (see S1 Table). The highest CI was obtained for the near future at Bergamo (0.27˚C) and for the far future at Valletta (1.76˚). The highest CI values observed for MENA cities are at least 2 times higher than those obtained by Guerreiro et al., (2018) [1] for both future periods. In addition, around 62% of the MENA cities will have a future CI greater than the highest one calculated for European cities.

Taking into account the importance of the projected changes, their consequences on cities with high population growth rates are uncertain. Coffel et al., (2017) [66] assessed the heat stress exposure during the 21st century. They found that population exposure to wet bulb temperatures linked to heat waves will increase by a factor of five to ten. They also highlighted the special situation of Africa due to their low adaptive capacity and rapidly growing population. More recently, Rohat et al., (2019) [67] conducted a study on the projections of human exposure to dangerous heat in some African cities under different scenarios. They found that most of the African cities will face significantly increased exposure to dangerous heat due to their location in hot regions and the high rates of urban population growth. In particular, they observed that the exposure in African cities will increase 20–52 times by the 2090s, depending on the scenario, being specially affected those cities in the Western and Eastern Africa. They raised the importance of the shift from a high to low urban population growth to reduce heat exposure and to address adaptation measures. For that purpose, the present work can help to identify those cities with a greater risk of suffering heat waves and its intensity (Table 2) to conduct adaptation and mitigation policies.

In this sense, previous works have focused their attention on the importance of adopting mitigation and adaptation measures and policies in the face of extreme events. Most of these studies were carried out in well-developed countries that belong to the first world [68–71]. However, it is crucial to focus on less developed countries and emergent economies [72–75]. The general conclusion of most of these articles is the importance of carrying out immediate actions to adapt to, and mitigate future climate changes. However, these studies also emphasize on the difficulty and cost for implementing these actions. If the challenge is extraordinary for most of the first world countries, it poses an almost unsolvable problem for some of the countries with the lowest GDPs worldwide. The aridity, lack of vegetation, lack of rainfall or

**Table 2. Top 5 cities for each variable: Maximum temperature (Tmax), percentage of heat waves days (HWD), mean (<I>) and maximum (MI) intensity, and cumulative index (CI).**

| Tmax | HWD | <I> | MI | CI |
|---|---|---|---|---|
| Baghdad | Mogadishu | Riyadh | Alexandria | Riyadh |
| Riyadh | Freetown | Yerevan | Giza | Tehran |
| Kuwait City | Riyadh | Tbilisi | Istanbul | Addis Ababa |
| Abu Dabhi | Lagos | Tehran | Ankara | Sanaa |
| Khartoum | Accra | Ankara | Yerevan | Yerevan |

extreme temperatures make these areas the auspicious scenario for a humanitarian catastrophe. Only those countries situated in the Persian Gulf have the capability and the resources to mitigate the impact of future climate changes.

## 5. Conclusions

The present work focuses on the vulnerability of the most populated cities of the MENA region to severe heat waves in the future. The main conclusions can be summarized as:

- Maximum temperature is expected to rise throughout the entire MENA region for the 21st century under the RCP 4.5 and 8.5 scenarios. The increment of maximum temperatures is expected to range between 4°C and 7°C depending of the area, being the most affected cities those situated in the Arabian Peninsula and the less those in the Gulf of Guinesla.

- As a consequence of the temperatures growth, more heat wave days are also expected. Focusing on the far future (2070–2099) under the RCP8.5 scenario, the 80% of the capitals from the MENA region will be more than 50% of days under heat wave conditions and, approximately, 50% of them will be more than 70% of days under heat wave conditions. In particular, there are some cities as Accra (Ghana), Lome (Togo), Mogadishu (Somalia) or Riyadh (Saudi Arabia) that will be practically the entire warm seasons under heat wave conditions.

- The mean and maximum intensity of heat waves is also expected to increase for almost the whole MENA region. Focusing on the far future under the RCP8.5, mean heat waves intensity will growth between 2°C and 4°C depending on the area, being the most affected cities those situated in the Middle East, eastern Africa and the Mediterranean area and the less those in the Gulf of Guinea. Additionally, maximum heat wave intensity will exceed the threshold value in more than 12°C in cities as Alexandria, Giza or Istanbul.

- The fact of the duration and intensity of the heat waves are not positively correlated has suggested the use of a cumulative index to determine the most threatened areas in the future. Results show that the Middle East area and the Eastern part the Africa will suffer the least favorable conditions according to this index, both for the near and far future.

The present work aims to alert of the disastrous situation that may take place during the current century due to climate change, especially in Africa. The increase of temperatures and, consequently, of heat waves even under a moderate scenario can lead to severe impacts in some of the least developed regions of the MENA domain. In fact, the results obtained suggest that at the end of the century, some of the African countries will be affected by heat wave conditions during almost the entire warm season. The special properties of the area and the lack of resources are additional difficulties to adapt and mitigate the impacts of climate change over the region.

## Supporting information

**S1 Table. List of models from CORDEX.**
(DOCX)

**S2 Table. Maximum temperature (value ± SD) in °C averaged over the periods 1970–1999, 2020–2049 and 2070–2099.** RCP4.5 and RCP8.5 refer to the two representative concentration pathways used during the calculations. *All the results are statistically significant at more than 95%.*
(DOCX)

**S3 Table. Percentage of days (value ± SD) under extreme temperature conditions per month and averaged over the periods 1970–1999, 2020–2049 and 2070–2099.** RCP4.5 and RCP8.5 refer to the two representative concentration pathways used during the calculations. *All the results are statistically significant at more than 95%.*
(DOCX)

**S4 Table. Percentage of days (value ± SD) under heat wave conditions per month and averaged over the periods 1970–1999, 2020–2049 and 2070–2099.** RCP4.5 and RCP8.5 refer to the two representative concentration pathways used during the calculations. *All the results are statistically significant at more than 95%.*
(DOCX)

**S5 Table. Mean and maximum intensity of days (value ± SD) under heat wave conditions and averaged over the periods 1970–1999, 2020–2049 and 2070–2099.** RCP4.5 and RCP8.5 refer to the two representative concentration pathways used during the calculations. *All the results are statistically significant at more than 95%.*
(DOCX)

**S6 Table. Cumulative Index for the periods 2020–2049 and 2070–2099.** RCP4.5 and RCP8.5 refer to the two representative concentration pathways used during the calculations. *All the results are statistically significant at more than 95%.*
(DOCX)

**S1 Fig.** Cluster analysis of the percentages of heat wave days for near future (2020–2049) under the **a)** RCP45 scenario; **b)** RCP8.5 scenario. Blue color for low values and red color for high values.
(TIF)

**S2 Fig.** Cluster analysis of the mean (left panels) and maximum (right panels) heat wave intensity for near future (2020–2049) under the **a, b)** RCP45 scenario; **c, d)** RCP8.5 scenario. Blue color for low values and red color for high values.
(TIF)

## Author Contributions

**Conceptualization:** R. Varela, M. deCastro.

**Data curation:** R. Varela, L. Rodríguez-Díaz.

**Formal analysis:** R. Varela, L. Rodríguez-Díaz.

**Funding acquisition:** M. deCastro.

**Investigation:** R. Varela, L. Rodríguez-Díaz, M. deCastro.

**Methodology:** R. Varela, L. Rodríguez-Díaz, M. deCastro.

**Resources:** M. deCastro.

**Software:** R. Varela, M. deCastro.

**Supervision:** M. deCastro.

**Validation:** R. Varela, L. Rodríguez-Díaz.

**Writing – original draft:** R. Varela.

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
