## [Decision Letter · Decision Letter 0]

1 Sep 2020

PONE-D-20-22188

Persistent heat waves projected for Middle East and North Africa by the end of the 21st century

PLOS ONE

Dear Dr. Varela,

Thank you for submitting your manuscript to PLOS ONE. After careful consideration, we feel that it has merit but does not fully meet PLOS ONE’s publication criteria as it currently stands. Therefore, we invite you to submit a revised version of the manuscript that addresses the points raised during the review process.

Substantial revisions on this manuscript are required to: (1) make a comprehensively evaluate the models’ ability in capturing the heat wave features in historical climate, by comparing with observation dataset; (2) provide more evidences for supporting the conclusion of negative correction between the projected duration and intensity of heat waves. Furthermore, it would be more interesting to provide some (supplementary) figures for describing the projected future changes of heat wave features over the whole Middle East and North Africa.

We look forward to receiving your revised manuscript.

Kind regards,

Delei Li, Ph.D.

Academic Editor

PLOS ONE

Journal Requirements:

4. We note that Figures in your submission contain map images which may be copyrighted. All PLOS content is published under the Creative Commons Attribution License (CC BY 4.0), which means that the manuscript, images, and Supporting Information files will be freely available online, and any third party is permitted to access, download, copy, distribute, and use these materials in any way, even commercially, with proper attribution. For these reasons, we cannot publish previously copyrighted maps or satellite images created using proprietary data, such as Google software (Google Maps, Street View, and Earth). For more information, see our copyright guidelines: http://journals.plos.org/plosone/s/licenses-and-copyright.

4.1.    You may seek permission from the original copyright holder of Figures to publish the content specifically under the CC BY 4.0 license. 

4.2.    If you are unable to obtain permission from the original copyright holder to publish these figures under the CC BY 4.0 license or if the copyright holder’s requirements are incompatible with the CC BY 4.0 license, please either i) remove the figure or ii) supply a replacement figure that complies with the CC BY 4.0 license. Please check copyright information on all replacement figures and update the figure caption with source information. If applicable, please specify in the figure caption text when a figure is similar but not identical to the original image and is therefore for illustrative purposes only.

Reviewers' comments:

Reviewer's Responses to Questions

**Comments to the Author**

1. Is the manuscript technically sound, and do the data support the conclusions?

Reviewer #1: Partly

Reviewer #2: Yes

2. Has the statistical analysis been performed appropriately and rigorously? 

Reviewer #1: Yes

Reviewer #2: No

3. Have the authors made all data underlying the findings in their manuscript fully available?

Reviewer #1: Yes

Reviewer #2: Yes

4. Is the manuscript presented in an intelligible fashion and written in standard English?

Reviewer #1: No

Reviewer #2: Yes

5. Review Comments to the Author

Reviewer #1: This study has analyzed the duration and intensity of heat wave in the near-future and far-future for 53 cities in the Middle East and the North Africa (MENA) region for the 21st century. The output of 13 regional models were used to analyze the changes in extreme heat wave under two different scenarios (RCP4.5 and RCP8.5). Although I found the manuscript interesting, it needs revision. The introduction needs major revision.

- Merge two first paragraphs of the introduction into one.

- Introduction: Sentences are too long (e.g., 67-71) Make sentences shorted and more fluent.

- Line 29: Write numbers lower than 10 in letter, check it in all text

- Introduction needs more references for example line 33, 34, 44,

- Lina 35, Add the criterion to define the megacity so readers would have an idea of what you are talking about

- Line 39-43: Make sentences more fluent. For example: The required infrastructure services (e.g., clean water, electricity, transportation, etc) and the agglomeration of population, make cities highly vulnerable to climate change.

- Line 42: How the rapid increase of urbanization is increasing the number of people living in slums.

- Line 44: To say "On the other hand," you have to have "on one hand" phrase in previous sentences. Check it in the entire text.

- 54-59: The rise of temperature will lead to an increase in the frequency and persistent of heat waves in the future [References] with significant impact on human mortality [add reference here] and socio-economic conditions [references]

- Line 71, remove "in fact"

- Line 76 remove “as mentioned” and change the sentence to “The study area represent …”

- line 77 either add a reference or remove the sentence

- Line 78-80: modify the sentence as follow: In the past 20 years, MENA region showed the highest population gross in the world [reference]

- line 80 replace "on the other hand" with "besides"

- "Besides, the vast majority of MENA are characterized by frequent extreme temperature events (>35C) [add reference] and severe drought in summer [add reference], leading to large arid zones that may increase the risk of famine and famine-related mortality.

- line 89. change it to "Based on our knowledge now, "

- line 92-94. merge this sentence with the next sentence in line 95-96

- line 101. what near surface means?

- Data and method:

- How many grid-cell did you analyzed for each city?

- Did you consider the entire area of the city or just one point (center point of the city)?

- line 128, remove the numbering.

- line 139 after 2-6 days add references s\\like:

- Mazdiyasni, O., Aghakouchak, A., 2015. Substantial increase in concurrent droughts and heatwaves in the United States. Proc. Natl. Acad. Sci. 112, 11484–11489. https://doi.org/10.1073/pnas.1422945112.

- Tavakol, Ameneh, Vahid Rahmani, and John Harrington Jr. "Evaluation of hot temperature extremes and heat waves in the Mississippi River Basin." Atmospheric Research 239 (2020): 104907. https://doi.org/10.1016/j.atmosres.2020.104907

- Nissan, H., Burkart, K., Coughlan de Perez, E., Van Aalst, M., Mason, S., 2017. Defining and predicting Heat Waves in Bangladesh. J. Appl. Meteorol. Climatol. 56, 2653–2670. https://doi.org/10.1175/JAMC-D-17-0035.1.

- line 165: explain the parameters

- Add a figure showing what months you have considered for analysis in each city?

- What the distribution of months for different location means? Discuss it.

- Except for the statistics of changes in the frequency and intensity of events, you have to specify which regions will experience extreme events in the future. Summarize the spatial occurrence of most significant changes.

- I suggest you use clustering methods (e.g., average linkage, k-mean, etc.) to cluster stations based on the frequency and intensity of events

- Is there a statistically significant difference between the changes in the frequency and intensity of events in near and far future?

- Line 342: higher than what?

- How the distribution of different months for each location influenced the results.

- Figures:

- Add unit to the figures’ legends

- Add scale to figures

- What HWD and IM means in table 1?

Reviewer #2: Review comments on “Persistent heat wave projected for Middle East and North Africa by the end of 21 century”

This study investigates the projected changes in heat waves in the Middle East and the North Africa (MENA) region under RCP4.5 and RCP8.5 scenarios using multi RCMs with CORDEX frame. Results show that 80% of the most populated MENA cities are expected to be at least 50% of the days under heat wave conditions during the warm season by the end of the century. Interesting, it’s revealed that changes in the duration and intensity of heat waves have shown to be negatively correlated. And Middle East and the eastern part of Africa will suffer the most unfavorable temperature conditions in the future. However, currently, it has evident limits, which could be improved considering the following comments. I suggest the authors to resubmit the manuscript after substantial revisions.

Major Comments:

1. Firstly, it is not enough verified that the duration and intensity of heat waves is negatively correlated. To provide a spatial distribution of correlation between duration and intensity series of heat waves may help.

2. Secondly, the conclusions are not surprising to me. And I can only consider the fourth one of innovation.

3. There are many integral indices that consider both the duration and intensity of heat waves. For example, Wang et al., 2019. Wang, P., Hui, P., Xue, D., & Tang, J. (2019). Future projection of heat waves over China under global warming within the CORDEX-EA-II project. Climate Dynamics, 53(1–2), 957–973. https://doi.org/10.1007/s00382-019-04621-7

Will the results be different with a different index?

4. Another concern is the authors stress the urbanization a lot in introduction part while do not in deeded connect the future projections with the increased population and specifically illustrate the threat. Such work can refer to Coffel et al. 2017.

Coffel, E. D., Horton, R. M., & de Sherbinin, A. (2017). Temperature and humidity based projections of a rapid rise in global heat stress exposure during the 21st century. Environmental Research Letters, 13(1), 14001.

5. How do the author choose the RCMs and the authors make no efforts to illustrate the intermodal spread among 13 models.

6. It seems that this study lacking the model validation for the historical climate.

7. Line 128-136. What is the difference between condition 1 and 2 ? It seems that condition 2 is true when condition 1 is true.

8. Lin 214-215 Lacking explanations on the regional differences.

Minor Comments:

Line 27-29: Not a clear expression.

Line 95: “determine” to be “predict”

Line 119 Does the “multi-model approach” means “multi-model ensemble mean”?

6. PLOS authors have the option to publish the peer review history of their article (what does this mean?). If published, this will include your full peer review and any attached files.

Reviewer #1: **Yes: **Ameneh Tavakol

Reviewer #2: No

---

## [Author Response · Author response to Decision Letter 0]

1 Oct 2020

Reviewer #1: This study has analyzed the duration and intensity of heat wave in the near-future and far-future for 53 cities in the Middle East and the North Africa (MENA) region for the 21st century. The output of 13 regional models were used to analyze the changes in extreme heat wave under two different scenarios (RCP4.5 and RCP8.5). Although I found the manuscript interesting, it needs revision. The introduction needs major revision.

- Merge two first paragraphs of the introduction into one.

This suggestion was considered in the new version of the manuscript

- Introduction: Sentences are too long (e.g., 67-71) Make sentences shorted and more fluent.

This suggestion was considered in the new version of the manuscript

- Line 29: Write numbers lower than 10 in letter, check it in all text

This suggestion was considered in the new version of the manuscript

- Introduction needs more references for example line 33, 34, 44,

This suggestion was considered in the new version of the manuscript

- Lina 35, Add the criterion to define the megacity so readers would have an idea of what you are talking about

This suggestion was considered in the new version of the manuscript

- Line 39-43: Make sentences more fluent. For example: The required infrastructure services (e.g., clean water, electricity, transportation, etc.) and the agglomeration of population, make cities highly vulnerable to climate change.

This suggestion was considered in the new version of the manuscript

- Line 42: How the rapid increase of urbanization is increasing the number of people living in slums.

Arimah and Branch (2011) conducted a study on the prevalence of slums in African countries. They linked the prominence given to urbanization as a major factor driving the proliferation of slums in Africa with the continent’s phenomenal urban transition. From 1950 to 2007 the African population living in urban areas grew from 14.5% to 38.7%. A direct consequence of this demographic change was the urbanization of poverty. This fact is linked to the movement of poverty from rural to urban areas (UN-HABITAT, 2003).

- Line 44: To say "On the other hand," you have to have "on one hand" phrase in previous sentences. Check it in the entire text.

This suggestion was considered in the new version of the manuscript

- 54-59: The rise of temperature will lead to an increase in the frequency and persistent of heat waves in the future [References] with significant impact on human mortality [add reference here] and socio-economic conditions [references]

This suggestion was considered in the new version of the manuscript

- Line 71, remove "in fact"

This suggestion was considered in the new version of the manuscript

- Line 76 remove “as mentioned” and change the sentence to “The study area represents …”

This suggestion was considered in the new version of the manuscript

- line 77 either add a reference or remove the sentence

This suggestion was considered in the new version of the manuscript

- Line 78-80: modify the sentence as follow: In the past 20 years, MENA region showed the highest population gross in the world [reference]

This suggestion was considered in the new version of the manuscript

- line 80 replace "on the other hand" with "besides"

This suggestion was considered in the new version of the manuscript

- "Besides, the vast majority of MENA are characterized by frequent extreme temperature events (>35C) [add reference] and severe drought in summer [add reference], leading to large arid zones that may increase the risk of famine and famine-related mortality.

This suggestion was considered in the new version of the manuscript

- line 89. change it to "Based on our knowledge now, "

This suggestion was considered in the new version of the manuscript

- line 92-94. merge this sentence with the next sentence in line 95-96

This suggestion was considered in the new version of the manuscript

- line 101. what near surface means?

Based on the CORDEX Archive Design, Near-Surface means at a height between 1.5 to 10.0 m (https://is-enes-data.github.io/cordex_archive_specifications.pdf)

- Data and method:

- How many grid-cell did you analyzed for each city?

- Did you consider the entire area of the city or just one point (center point of the city)?

The spatial resolution of the models used in the present study is 0.44º x 0.44º. To analyse the cities, first we selected the closest grid node to the city coordinates. Second, a bilinear interpolation was performed, considering the closest four grid nodes to the city coordinate (stablished in the first step), to determine the maximum temperature.

Action: This information is now clarified in the method section (Lines 112-114).

- line 128, remove the numbering.

This suggestion was considered in the new version of the manuscript

- line 139 after 2-6 days add references s\\like:

This suggestion was considered in the new version of the manuscript

- line 165: explain the parameters

Parameters were explained in lines 168-170. 

- Add a figure showing what months you have considered for analysis in each city?

The months considered for each city are shown in Table S2 of the Supplementary Information. Therefore, the authors consider that this figure unnecessarily enlarge the document. In any case, if the reviewer considers that a figure showing the months is essentia, we could show a figure of the type shown below:

- What the distribution of months for different location means? Discuss it.

Those months whose mean maximum temperature exceeds the 75th percentile, were selected as part of the warm season. The use of this threshold is due to the particular characteristics of the MENA region. For example, those cities closest to the equator show hardly any seasonality while others show a clear season of warmer temperatures. Thus, the warm season is different for each city.

Previous articles usually used the same warm season for the whole area under study because they were focused on regions with similar conditions. In our case, it is mandatory to establish different months to assess the particular characteristics of each city. Selecting the same period for all zones could induce errors when using the maximum temperatures.

Action: This information is now clarified in the method section (Lines 119-122).

- Except for the statistics of changes in the frequency and intensity of events, you have to specify which regions will experience extreme events in the future. Summarize the spatial occurrence of most significant changes.

- I suggest you use clustering methods (e.g., average linkage, k-mean, etc.) to cluster stations based on the frequency and intensity of events.

We want to acknowledge the reviewer's suggestion that led us to use a clustering method to assess the spatial differences in the MENA region.

We have introduced two new figures (Figures 5 and 8) showing the cluster analysis for the percentage of heat wave days and the mean and maximum intensity. These figures clearly show the spatial differences and the negative correlation between the heat wave days and the mean and maximum intensity.

Moreover, the new Table 2 summarizes the most affected cities in terms of temperature, percentage of heat wave days, mean and maximum intensity and cumulative index.

Action: This information is now clarified in the results section (Lines 259-266 and 290-297) and in the discussion section (Lines 328-336). New Figures 5 and 8 and a new Table 2 have been added to the manuscript. In addition, new Figures S1 and S2 have been add to the Supplementary information section.

- Is there a statistically significant difference between the changes in the frequency and intensity of events in near and far future?

All the results in the Supplementary Information section are statistically significant at more than 95%. We did not observe any statistically significant difference between near and far future.

Action: This information is now clarified in the captions of the Supplementary information.

- Line 342: higher than what?

This suggestion was considered in the new version of the manuscript

- How the distribution of different months for each location influenced the results.

The main objective of the present study is focused on the extreme events which are reached in those months with higher temperatures (warm season). Due to the particular characteristics of the MENA region, different warm seasons were obtained depending of the city. 

Evidently, a change in the choice of the warm season could affect the results. The use of the 95th percentile only in those months of interest is a very restrictive condition to assess the most extreme events. Thus, selecting more months could change the threshold value and the different heat waves parameters.

This method has been previously applied by different authors as, for example, Lelieveld et al., (2016) who studied the warm season from June to August. The main difference with the present study is that our area is wider and the warm season differs between the cities. 

- Figures:

- Add unit to the figures’ legends

- Add scale to figures

These suggestions were considered in the new version of the manuscript

- What HWD and IM means in table 1?

HWD means Heat Wave Days and IM was changed by MI and refers to de maximum intensity. 

Action: This information is now clarified in the caption of the Table 1.

- References:

Arimah, B. C., & Branch, C. M. (2011). Slums as expressions of social exclusion: Explaining the prevalence of slums in African countries. UN-Habitat, Nairobi.

Lelieveld, J., Proestos, Y., Hadjinicolaou, P., Tanarhte, M., Tyrlis, E. and Zittis, G. (2016). Strongly increasing heat extremes in the Middle East and North Africa (MENA) in the 21st century. Clim. Change 137(1-2), 245-260.

UN–HABITAT (2003) Guide to Monitoring Target 11: Improving the Lives of 100 million Slum Dwellers: Progress towards the Millennium Development Goals, UN-HABITAT, Nairobi.

Reviewer #2: Review comments on “Persistent heat wave projected for Middle East and North Africa by the end of 21 century”

This study investigates the projected changes in heat waves in the Middle East and the North Africa (MENA) region under RCP4.5 and RCP8.5 scenarios using multi RCMs with CORDEX frame. Results show that 80% of the most populated MENA cities are expected to be at least 50% of the days under heat wave conditions during the warm season by the end of the century. Interesting, it’s revealed that changes in the duration and intensity of heat waves have shown to be negatively correlated. And Middle East and the eastern part of Africa will suffer the most unfavorable temperature conditions in the future. However, currently, it has evident limits, which could be improved considering the following comments. I suggest the authors to resubmit the manuscript after substantial revisions.

Major Comments:

1. Firstly, it is not enough verified that the duration and intensity of heat waves is negatively correlated. To provide a spatial distribution of correlation between duration and intensity series of heat waves may help.

The negative correlation between duration and intensity was obtained following the process described below:

 We calculate the annual average heat wave days, mean and maximum intensity during the warm season of each city over the period of 30 years for the near and far future.

 We represent Figures 4, 6 and 7 with these data.

In general, those regions with the highest intensity of heat waves are the ones with the shortest duration (and vice versa), which lead us to think that both parameters are anti-correlated. A Spearman test has been used to calculate this anti-correlation. The results of this test can be observed in Table 1 of the manuscript.

Furthermore, the reviewer can observe these anti-correlations in the figures shown below.

Figure 1: Correlation between the percentage of heat wave days (HWD) and the mean (<I>, left panels) and maximum intensity (MI, right panels). Black line represents the linear trend and r values indicate the Spearman's rank correlation coefficient. a) and b) near future (2020-2049) under the RCP4.5 scenario; c) and d) far future (2070-2099) under the RCP4.5 scenario; e) and f) near future under the RCP8.5 scenario; g) and h) far future under the RCP8.5 scenario.

If the reviewer considers that these figures must be included in the Supplementary Information section, they would be added.

Moreover, new Figures 5 and 8 show the cluster analysis for the percentages of heat wave days and the mean and maximum intensity. Comparing both analysis, the highest (lowest) values for the percentage of heat wave days were obtained for the southern (northern) latitudes. However, for the mean and maximum intensity, the highest (lowest) values were obtained for the northern (southern) values. These results corroborate the negative correlation between the duration and the intensity. 

2. Secondly, the conclusions are not surprising to me. And I can only consider the fourth one of innovation.

The first three points of this section were focused on the results obtained for the evolution of extreme temperatures, heat wave days, and mean and maximum intensities. All these variables are expected to increase independently of the period or the scenario under study.

Previous studies focused on the MENA region showed similar results as in the present work. However, the great majority of these works were conducted from a more macroscopic point of view and focusing on sub domains (Lelieveld et al., 2016; Dosio, 2017; Bucchignani et al., 2018; Ozturk et al., 2018). Our objective was to focus on the particular characteristics of each city which differentiates us from previous articles. Moreover, has allowed us to show more detailed results for each area. Giugni et al., (2015) or Pal and Eltahir, (2016) conducted similar studies as in the present work, however they focused on particular cities of Africa and Asia, respectively but did not cover the entire area of MENA. Thus, beyond being expected results, we consider that we provide information different from the existing one.

Finally, the fourth conclusion deepens on the negative correlation between duration and intensity which leads us to use the Cumulative Index that shows regional differences together with Figures 5, 8 and 9.

3. There are many integral indices that consider both the duration and intensity of heat waves. For example, Wang et al., 2019. Wang, P., Hui, P., Xue, D., & Tang, J. (2019). Future projection of heat waves over China under global warming within the CORDEX-EA-II project. Climate Dynamics, 53(1–2), 957–973. https://doi.org/10.1007/s00382-019-04621-7

Will the results be different with a different index?

Since the first studies focused on extreme events, different indices have been developed and used (Russo and Sterl, 2011; Perkins and Alexander, 2012; Russo et al., 2014, 2015; Dosio, 2017; Wang et al., 2019). Most of these indices have been based on the heat wave duration, intensity or both such as the recently developed Heat Wave Magnitude index (HWMI) (Russo et al., 2015). In our article, we evaluate the duration and intensity, the indices that define a heat wave. Moreover, in a similar way as previous articles, we also used an index to consider both parameters at the same time to compare the different regions under study. With this method, we could assess the regional differences taking into account duration and intensity. 

It is evident that the use of different definitions or indices may produce some variations in the results. For example, an important point is the definition of a heat wave. In the present article we defined a heat wave as a minimum of 3 consecutive days under extreme temperature conditions exceeding a temperature threshold (95th percentile of the daily maximum temperature centered on a 31-day window) (deCastro et al., 2011; Russo et al., 2014; Dosio, 2017). However, previous articles also used other definitions like a duration from 2 to 6 days or different percentiles (90th, 99th…). These changes in the definition of heat waves or indices lead to obtaining different results. But, the most important thing to consider is that, beyond the definitions or indices used in any article, all these previous studies always come to similar conclusions: the temperature will continue to increase (Evans, 2009; Niang et al., 2014; Pal and Eltahir, 2016; Bucchignani et al., 2018; Ozturk et al., 2018) and thus, episodes of unusual extreme temperatures and heat waves are expected to rise in the future. Moreover, both the duration and intensity of heat waves are expected to continue growing until the end of the century (Lelieveld et al., 2016; Dosio, 2017; Ozturk et al., 2018).

4. Another concern is the authors stress the urbanization a lot in introduction part while do not in deeded connect the future projections with the increased population and specifically illustrate the threat. Such work can refer to Coffel et al. 2017.

Coffel, E. D., Horton, R. M., & de Sherbinin, A. (2017). Temperature and humidity based projections of a rapid rise in global heat stress exposure during the 21st century. Environmental Research Letters, 13(1), 14001.

The main reason for giving so much importance to the population and urbanization of MENA is due to the special characteristics of these areas compared to other regions of the world. In the MENA region, a continued population growth is expected, which, together with some poor living conditions and rising temperatures, can have unprecedented humanitarian consequences.

In any case, the reviewer is totally right and we have dedicated a paragraph in the discussion section of the new version of the manuscript to link the increase in temperatures and urbanization.

Action: The discussion section has been improved to address the connections between increasing temperatures and urbanization. Lines 372-386.

5. How do the author choose the RCMs and the authors make no efforts to illustrate the intermodal spread among 13 models.

The choice of the RCMs was due to the availability of the models at https://www.cordex.org/data-access/ for the Africa domain. We have used all the models at our disposal that met the parameters shown in the methods section.

In addition, the following figure shows the Tmax anomaly averaged for the entire study area for each RCM, future period, and greenhouse gas emission scenario. The shadowed area of this figure shows the intermodal dispersion between the RCMs used in this study.

Figure 2: Average at all models of the Tmax anomaly (black line) +2SD (shadow area). The colored lines represent the Tmax anomaly of each RCM of the 13 CORDEX models. a) near future (2020-2049) under the RCP4.5 scenario; b) far future (2070-2099) under the RCP4.5 scenario; c) near future under the RCP8.5 scenario; d) far future under the RCP8.5 scenario.

As can be observed in the Figure 2, all RCMs show an increasing trend for any period and scenario. Moreover, the Tmax anomaly for all models are within the mean value (black line) ± 2 standard deviations (SD). Therefore, all the RCMs can be considered to perform the multi-model mean.

6. It seems that this study lacking the model validation for the historical climate.

The skill of RCMs from CORDEX project to reproduce historical data in the area under study was evaluated in previous studies (Russo et al., 2016; Dosio, 2017; Ozturk et al., 2018). Russo et al., (2016), Haensler et al., 2013; Laprise et al., 2013; Buontempo et al., 2014; Giorgi et al., 2014; Dosio, 2017; Ozturk et al., 2018). These studies used different RCMs driven by different GCMs to analyze both historical data and future projections. In particular, Kim et al., (2013) conducted an evaluation of TAVG, TMAX, TMIN, and other variables in the CORDEX-Africa hindcast, using a variety of reference datasets and comparison of outputs from multiple models against multiple observations. They showed that RCM skill varies according to variables, regions, and metrics being generally better for TAVG and TMAX. Russo et al., (2016) carried out a model evaluation for the historical period (1979-2015) by means of an ensemble of 13 RCMs of the CORDEX-Africa multi-model scenario experiment with a resolution of 0.44° (similar to the present article). They based their assessment on how realistic the ensemble models could reproduce the magnitude and extent of HWMId for each season. They obtained that the ensemble of all models at different HWMId levels captures the general trend of heat waves in the spatial area during the period 1979–2015. Taking into account that the models used in the present study have been validated in previous works, we considered it sufficient to refer to those studies and not make the manuscript longer. For a more complete description, the reviewer is referred to Russo et al., (2016) and Dosio (2017).

Action: This information is now clarified in the methods section. Lines 103-106.

7. Line 128-136. What is the difference between condition 1 and 2? It seems that condition 2 is true when condition 1 is true.

The second condition applies to the models that met the first condition. Models that have met both conditions will be deemed to meet the consensus criterion.

The consensus criterion was previously used in several studies (Pfeifer et al., 2015; Koletsis et al., 2016; Costoya et al., 2019). 

〖∆T〗_max was calculated for the 13 RCMs and the multi-model mean. Then, the first condition was applied. Thus, for every pixel, at least 60% of the models (in our case it means at least 8 models), must have the same sign in the trend as the multi-model mean.

Then the second condition was applied at every pixel to obtain its significance level. To achieve this condition, the pixel should present a 5% significance level for at least 60% of those 8 models that fulfilled the first condition (in our case it means at least 5 models).

8. Line 214-215 Lacking explanations on the regional differences.

In order to evaluate the regional differences, a clustering method have been carried out in the results section (Lines 260-267 and 290-299) and discussed in the discussion section (Lines 330-339). New Figures 5 and 8 and Table 2 have been added to the Manuscript. New Figures S1 and S2 have been added to the Supplementary Information section.

Minor Comments:

Line 27-29: Not a clear expression.

This suggestion was considered in the new version of the manuscript

Line 95: “determine” to be “predict”

This suggestion was considered in the new version of the manuscript

Line 119 Does the “multi-model approach” means “multi-model ensemble mean”?

Yes, we considered the ensemble of the models under study.

- References:

Bucchignani, E., Mercogliano, P., Panitz, H. J. and Montesarchio, M. (2018). Climate change projections for the Middle East–North Africa domain with COSMO-CLM at different spatial resolutions. Adv. Clim. Change Res. 9(1), 66-80.

Buontempo C, Mathison C, Jones R, Williams K, Wang C, McSweeney C (2014). An ensemble climate projection for Africa. Clim Dyn.

Costoya, X., deCastro, M., Santos, F., Sousa, M. C. and Gomez-Gesteira, M. (2019). Projections of wind energy resources in the Caribbean for the 21st century. Energy 178, 356-367.

deCastro, M., Gomez-Gesteira, M., Ramos, A. M., Alvarez, I. and deCastro, P. (2011). Effects of heat waves on human mortality, Galicia, Spain. Clim. Res. 48(2-3), 333-341.

Dosio, A. (2017). Projection of temperature and heat waves for Africa with an ensemble of CORDEX Regional Climate Models. Clim. Dyn. 49(1-2), 493-519.

Evans, J. P. (2009). 21st century climate change in the Middle East. Clim. Change 92 (3–4), 417–432.

Giorgi F, Coppola E, Raffaele F, Diro GT, Fuentes-Franco R, Giuliani G, Mamgain A, Llopart MP, Mariotti L, Torma C (2014). Changes in extremes and hydroclimatic regimes in the CREMA ensemble projections. Clim Change pp 39–51.

Giugni, M., Simonis, I., Bucchignani, E., Capuano, P., De Paola, F., Engelbrecht, F., Mercogliano, P. and Topa, M. E. (2015). The impacts of climate change on African cities. In Urban vulnerability and climate change in Africa (pp. 37-75). Springer, Cham.

Haensler A, Saeed F, Jacob D (2013). Assessing the robustness of projected precipitation changes over central Africa on the basis of a multitude of global and regional climate projections. Clim Change 121(2):349–363.

Kim J, Waliser DE, Ca Mattmann, Goodale CE, Hart AF, Pa Zimdars, Crichton DJ, Jones C, Nikulin G, Hewitson B, Jack C, Lennard C, Favre A (2013). Evaluation of the CORDEX-Africa multi-RCM hindcast: systematic model errors. Clim Dyn 42(5– 6):1189–1202.

Koletsis, I., Kotroni, V., Lagouvardos, K. and Soukissian, T. (2016). Assessment of offshore wind speed and power potential over the Mediterranean and the Black Seas under future climate changes. Renew. Sust. Energ. Rev. 60, 234-245.

Laprise R, Hernández-Díaz L, Tete K, Sushama L, Šeparović L, Martynov A, Winger K, Valin M (2013). Climate projections over CORDEX Africa domain using the fifth-generation Canadian Regional Climate Model (CRCM5). Clim Dyn.

Lelieveld, J., Proestos, Y., Hadjinicolaou, P., Tanarhte, M., Tyrlis, E. and Zittis, G. (2016). Strongly increasing heat extremes in the Middle East and North Africa (MENA) in the 21st century. Clim. Change 137(1-2), 245-260.

Niang, I., Ruppel, O. C., Abdrabo, M. A., Essel, A., Lennard, C., Padgham, J. and Urquhart, P. (2014). Africa Climate Change 2014: Impacts, Adaptation, and Vulnerability. Part B: Regional Aspects. Contribution of Working Group II to the Fifth Assessment Report of the Intergovernmental Panel on Climate Change ed VR Barros et al.

Ozturk, T., Turp, M. T., Türkeş, M. and Kurnaz, M. L. (2018). Future projections of temperature and precipitation climatology for CORDEX-MENA domain using RegCM4.4. Atmos. Res. 206, 87-107.

Pal, J. S. and Eltahir, E. A. (2016). Future temperature in southwest Asia projected to exceed a threshold for human adaptability. Nat. Clim. Change 6(2), 197.

Perkins, S. E. and Alexander, L. V. (2012). On the measurement of heat waves, J. Clim. 26, 4500–4517.

Pfeifer, S., Bülow, K., Gobiet, A., Hänsler, A., Mudelsee, M., Otto, J., ... & Jacob, D. (2015). Robustness of ensemble climate projections analyzed with climate signal maps: seasonal and extreme precipitation for Germany. Atmosphere, 6(5), 677-698.

Russo, S. and Sterl, A. (2011). Global changes in indices describing moderate temperature extremes from the daily output of a climate model, J. Geophys. Res. 116.

Russo, S., Dosio, A., Graversen, R. G., Sillmann, J., Carrao, H., Dunbar, M. B., Singleton, A., Montagna, P., Barbola, P. and Vogt, J. V. (2014). Magnitude of extreme heat waves in present climate and their projection in a warming world. J. Geophys. Res.: Atmos. 119(22), 12-500. 

Russo, S., Sillmann, J. and Fischer, E. M. (2015). Top ten European heatwaves since 1950 and their occurrence in the coming decades. Environ. Res. Lett. 10(12), 124003.

Russo, S., Marchese, A. F., Sillmann, J. and Imme, G. (2016). When will unusual heat waves become normal in a warming Africa?. Environ. Res. Lett. 11(5), 054016.

Wang, P., Hui, P., Xue, D. and Tang, J. (2019). Future projection of heat waves over China under global warming within the CORDEX-EA-II project. Clim. Dyn. 53(1-2), 957-973.

---

## [Decision Letter · Decision Letter 1]

4 Nov 2020

Persistent heat waves projected for Middle East and North Africa by the end of the 21st century

PONE-D-20-22188R1

Dear Dr. Varela,

We’re pleased to inform you that your manuscript has been judged scientifically suitable for publication and will be formally accepted for publication once it meets all outstanding technical requirements.

Kind regards,

Delei Li, Ph.D.

Academic Editor

PLOS ONE

Additional Editor Comments (optional):

Reviewers' comments:

Reviewer's Responses to Questions

**Comments to the Author**

1. If the authors have adequately addressed your comments raised in a previous round of review and you feel that this manuscript is now acceptable for publication, you may indicate that here to bypass the “Comments to the Author” section, enter your conflict of interest statement in the “Confidential to Editor” section, and submit your "Accept" recommendation.

Reviewer #1: All comments have been addressed

2. Is the manuscript technically sound, and do the data support the conclusions?

Reviewer #1: Yes

3. Has the statistical analysis been performed appropriately and rigorously? 

Reviewer #1: Yes

4. Have the authors made all data underlying the findings in their manuscript fully available?

Reviewer #1: Yes

5. Is the manuscript presented in an intelligible fashion and written in standard English?

Reviewer #1: Yes

6. Review Comments to the Author

Reviewer #1: The manuscript "Persistent heat waves projected for Middle East and North Africa by the end of the 21st

century" has been improved significantly. All questions have been answered and the text has been improved.

7. PLOS authors have the option to publish the peer review history of their article (what does this mean?). If published, this will include your full peer review and any attached files.

Reviewer #1: No

---

## [Editor Report · Acceptance letter]

9 Nov 2020

PONE-D-20-22188R1 

Persistent heat waves projected for Middle East and North Africa by the end of the 21^st^ century 

Dear Dr. Varela:

I'm pleased to inform you that your manuscript has been deemed suitable for publication in PLOS ONE. Congratulations! Your manuscript is now with our production department. 

Kind regards, 

on behalf of

Dr. Delei Li 

Academic Editor

PLOS ONE